# Quantifying Uncertainty in Exposure to Coastal Hazards Associated with Both Climate Change and Adaptation Strategies: A U.S. Pacific Northwest Alternative Coastal Futures Analysis

**Alexis K. Mills [1,2], Peter Ruggiero [3,]*, John P. Bolte [1], Katherine A. Serafin [3,4] and Eva Lipiec [3]**

[1] Biological and Ecological Engineering, Oregon State University, Corvallis, OR 97331, USA; alexis.k.mills@gmail.com (A.K.M.); John.Bolte@oregonstate.edu (J.P.B.)
[2] Northwestern Division, US Army Corps of Engineers, Portland, OR 97232, USA
[3] College of Earth, Ocean, and Atmospheric Sciences, Oregon State University, Corvallis, OR 97331, USA; kserafin@ufl.edu (K.A.S.); eva.lipiec@gmail.com (E.L.)
[4] Geography, University of Florida, Gainesville, FL 32611, USA
[*] Correspondence: pruggier@coas.oregonstate.edu; Tel.: +1-541-737-1239

**Abstract:** Coastal communities face heightened risk to coastal flooding and erosion hazards due to sea-level rise, changing storminess patterns, and evolving human development pressures. Incorporating uncertainty associated with both climate change and the range of possible adaptation measures is essential for projecting the evolving exposure to coastal flooding and erosion, as well as associated community vulnerability through time. A spatially explicit agent-based modeling platform, that provides a scenario-based framework for examining interactions between human and natural systems across a landscape, was used in Tillamook County, OR (USA) to explore strategies that may reduce exposure to coastal hazards within the context of climate change. Probabilistic simulations of extreme water levels were used to assess the impacts of variable projections of sea-level rise and storminess both as individual climate drivers and under a range of integrated climate change scenarios through the end of the century. Additionally, policy drivers, modeled both as individual management decisions and as policies integrated within adaptation scenarios, captured variability in possible human response to increased hazards risk. The relative contribution of variability and uncertainty from both climate change and policy decisions was quantified using three stakeholder relevant landscape performance metrics related to flooding, erosion, and recreational beach accessibility. In general, policy decisions introduced greater variability and uncertainty to the impacts of coastal hazards than climate change uncertainty. Quantifying uncertainty across a suite of coproduced performance metrics can help determine the relative impact of management decisions on the adaptive capacity of communities under future climate scenarios.

**Keywords:** coastal hazards exposure; alternative futures analysis; climate change; Envision; coastal erosion; coastal flooding; Tillamook County; Oregon

## 1. Introduction

The inherent variability of dynamic coastal systems, combined with the pressure of coastal development, creates a high degree of uncertainty surrounding future coastal community sustainability [1–3]. The formulation of adaptation pathways in response to climate change poses challenges as it forces decision-making under unknown future conditions [4,5]. Policy-makers need ways of assessing the possible consequences of a range of decisions to accommodate and protect increasing populations along the coastline [6]. Understanding how to cope explicitly with uncertainty in rates and magnitudes of climate change and human adaptive response, as well as how propagated uncertainty may

limit the ability to quantify outcomes at a range of geographic scales, is important in the development of adaptive capacity within coastal communities.

Analyses and approaches that couple physical landscape processes related to climate change with human behavior have been used in vulnerability, mitigation, and adaptation studies (e.g., [7–9]). Index/indicator-based methods are among the most commonly applied approaches used to quantify coastal vulnerability due to their relative simplicity (e.g., [10–12]). Anfuso et al. [13] provide a recent review of these approaches, describing data needs and examples of vulnerability indices being employed in adaptation planning. While these approaches help identify aggregated relative risk along the shore based on past conditions, they are not often suitable for evaluating evolving forcing conditions or for assessing site specific variability (e.g., demographics are usually aggregated at the census block level and are updated every 10 years). Furthermore, index/indicator-based methods are limited in their ability to explore the feedbacks between natural and human systems, which are necessary to understand the potential outcomes of adaptation strategies. On the other hand, integrated scenario methodologies, which often combine storylines and simulation, have become state of the art in exploring socio-environmental changes [14] Integrated scenario analyses can provide important assessments of climate change and climate change policy, allowing analysts and stakeholders to explore complex and uncertain futures and address feedbacks between natural and human systems (e.g., [14–18]). However, combining climate change scenarios and human adaptation scenarios can be challenging as each type of scenario contains different forms of uncertainty. While climate change scenarios are predominately used to address uncertainty in physical systems, human adaptation scenarios are concerned with uncertainties in economic, social, political, and cultural systems [14].

In recent years, the U.S. Pacific Northwest (PNW) coast has seen a heightened risk of hazards because of relative sea-level rise (SLR) and changing storm frequency and intensity [19–23]. The underlying complexity of these phenomena complicates the prediction of future climate conditions at local scales (e.g., [24–26]). Recent forecasts of SLR by the end of the 21st century vary from approximately 0.1 to 1.5 meters along the PNW coast and are dependent upon vertical land motion, atmospheric, and cryospheric variables [24,27,28]. Furthermore, downscaled predictions of the evolving wave climate and the relative frequency and intensity of major El Niño Southern Oscillation (ENSO) events are also variable [29–32]. Thus, capturing coastal hazards exposure in response to fundamental uncertainty within each of these three climate drivers (SLR, wave climate, and ENSO frequency) and how that uncertainty may be exacerbated by their concurrence [33] is critical in designing robust adaptation strategies.

In addition to quantifying variability of the drivers of coastal hazards with respect to climate, informed adaptation also requires estimates of the uncertainty related to a range of possible human actions [4,6]. Responses to coastal hazards vary depending upon local social, political, and physical climate [34–37]. For example, communities in coastal areas may protect infrastructure through a range of solutions from structural engineering features (e.g., riprap revetments), to changes in land use (e.g., adjustments to zoning, urban and infrastructure development, and regulations; [38]), to managed retreat (e.g., [39]).

This paper evaluates the uncertainty of alternative coastal futures by combining a range of climate change and adaptation scenarios within our study site—Tillamook County, Oregon (Figure 1). In Tillamook County, approximately one quarter of all permanent residents live within a half mile of the Pacific Ocean and several communities already experience issues related to coastal flooding, erosion, and limited beach accessibility [40–43]. Community stakeholders are interested in exploring potential future coastal resilience, within the context of exposure to coastal hazards in a changing climate.

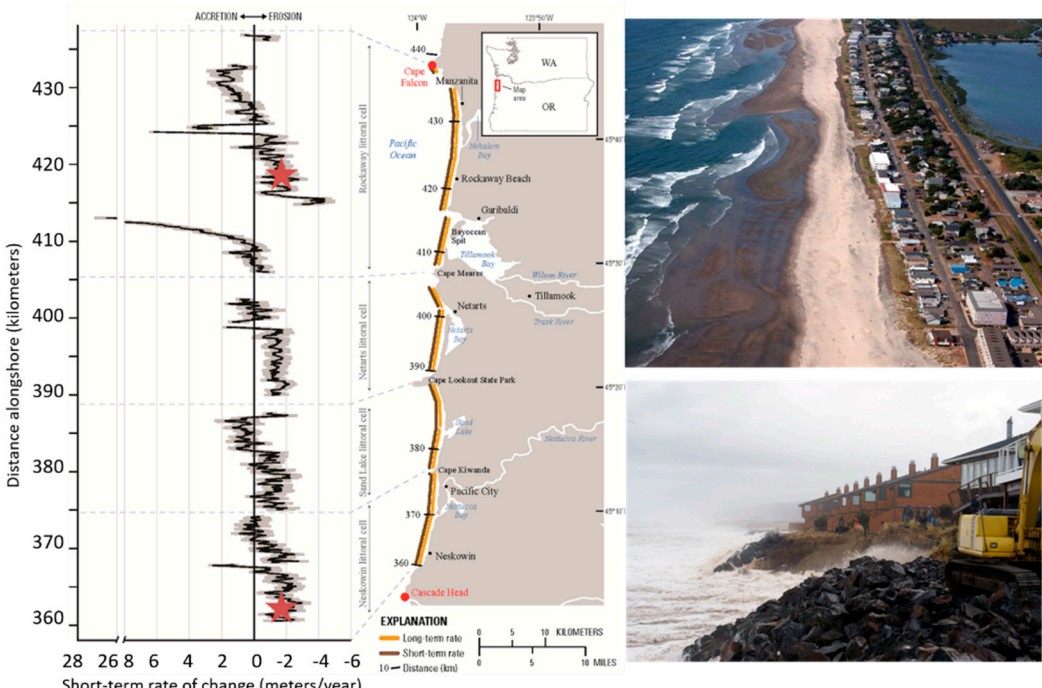

**Figure 1.** Left. Tillamook County, OR. Multidecadal shoreline change rates were computed between 1967 and 2002 (after [40]). Red stars represent the locations of the two photographs on the right (top—Rockaway Beach, OR and bottom—Neskowin, OR). Both communities are experiencing relatively high shoreline erosion rates (>1.0 m/year), are therefore exposed to coastal change and flood hazards, and have responded via the construction of riprap revetments.

A spatially explicit, policy-centric, agent-based modeling framework *Envision* [44], was used to examine interactions between human and natural systems across this county's shoreline. First, uncertainty was addressed in the form of individual climate drivers (i.e., wave height, sea-level rise) and human adaptation drivers (i.e., development restrictions, construction of backshore protection structures (BPS), such as riprap revetments). Second, uncertainty was examined within the context of integrated alternative future scenarios capturing both climate and management alternatives. Probabilistic simulations of total water levels (TWLs, [33]) along the shoreline captured the variability of sea-level rise, wave climate, and ENSO events under a range of climate change scenarios through the end of the twenty first century. In collaboration with a group of local stakeholders, the Tillamook County Coastal Knowledge-to-Action Network (KTAN) described in [42], a set of coproduced adaptation policy scenarios related to management decisions and socioeconomic trends were developed and used to explore variations in the human system. The KTAN included members from state, county, and local agencies, nongovernmental organizations, private citizens, researchers, students, and outreach specialists. This stakeholder network was interested in using Envision to evaluate how different adaptation policies and effects of climate change my impact coastal Tillamook County into the future. Using stakeholder defined landscape performance metrics related to coastal hazards exposure, the work described here explores two questions: (1) which human (policy) or physical drivers deviate the most from current (or baseline) conditions, and (2) how does hazard exposure uncertainty vary through time in response to human or physical drivers with respect to stakeholder defined landscape performance metrics?

## 2. Methods

The framework through which climate change, socioeconomic change, policy choices, and coastal hazards were simulated, as well as the methods for deriving climate and decision-making variability and uncertainty within that framework, are presented below.

### 2.1. Alternative Futuring through Coupled Human and Natural Systems Modeling

*Envision* [43,44] is a spatially explicit, multiagent-based modeling platform which couples biogeophysical models with socioeconomic drivers and management strategies to explore landscape change trajectories (Figure 2). This integrative modeling approach allows scientists and stakeholders to explore outcomes and tradeoffs that result when decision-making entities and their policies are included as part of evolving landscapes. *Envision* enables spatiotemporal simulation of landscape change through the synchronization of multiple submodels which are described below in the context of modeling regional scale (O(100 km)) coastal hazards in Tillamook County, OR (Figures 1 and 2) [42,43]. Mills et al. [43] provides a more detailed description of Envision, the coastal hazards submodels, and the simulation approach related to population growth and development. Most of the Tillamook County shoreline consists of sandy, dune-backed beaches, approximately 10% of which is backed by BPS. Almost half of the coast was eroding at rates exceeding one meter per year between 1967 and 2002 (Figure 1) [40].

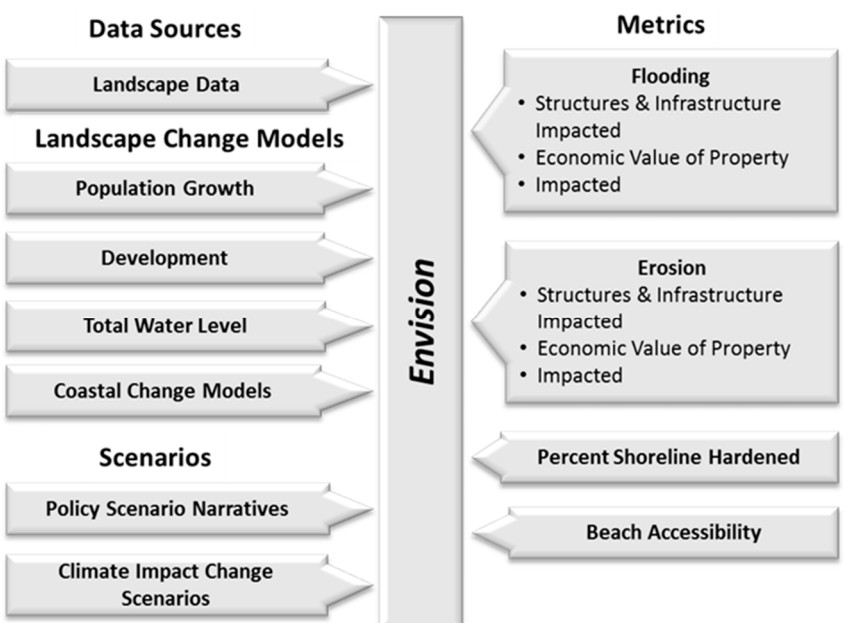

**Figure 2.** Envision inputs, landscape change models, and evaluative models specific to the modeling of coastal hazards in Tillamook County, Oregon from [43].

#### 2.1.1. Simulating Coastal Flood and Erosion Hazards

Within *Envision*, flooding due to storms, or event-based coastal flooding, was simulated using the maximum yearly extreme TWL relative to the elevation of relevant backshore features (e.g., dune crest) [25,40,45]. To derive the maximum yearly TWL, TWLs were calculated as a linear superposition of the tide, nontidal residual, and wave-induced runup [33,46] as follows:

$$TWL = MSL + \eta_A + \eta_{NTR} + R_{2\%} \tag{1}$$

where MSL is the mean sea level, $\eta_A$, is the deterministic astronomical tide, $\eta_{NTR}$, is the nontidal residual generated by physical processes including wind setup and barometric surge, and $R_{2\%}$, a function of wave height and wave length, is the two percent exceedance value of vertical wave runup on a beach or structure above the still water level [47]. The Total Water Level Full Simulation Model (TWL-FSM, [33]), was used to generate probabilistic time series of the components that comprise daily maximum TWLs, including wave height, wave period, wave direction, MSL, $\eta_A$, and $\eta_{NTR}$, capturing variability, nonstationarity, and the conditional dependencies in these parameters. The TWL-FSM simulates deep water wave conditions which were then transformed to the 20-m (m)

contour (or the seaward most location of the breaking parameter exceeding 0.4). A surrogate model (look-up table) of a finite library of SWAN (Simulating Waves Nearshore) model [48] simulations (details can be found in 33) was used to efficiently transform simulations of wave conditions [49].

The transformed waves were 'back-shoaled' using linear theory and the empirical relationship of [50] was used to compute $R_{2\%}$ on sandy dune backed beaches (most of the study area). Wave runup on BPS, bluffs, cliffs, and cobble berms was calculated based upon the Technical Advisory Committee for Water Retaining Structures (TAW) method, which provides a mechanism for adjusting the runup value based on parameters of the backshore feature (e.g., roughness, slope, and porosity; [49,51,52]). The coastal flooding hazard was assessed at 100 m alongshore resolution for the maximum yearly TWL event between 2010 and 2099 under varying climate scenarios, which are described below. Where the maximum yearly TWL exceeded the height of a backshore feature (i.e., dune toe or crest, BPS), the extent of flooding was computed using a simple bathtub model (e.g., [53]) allowing for the accounting of the amount of infrastructure impacted by these hazards.

In addition to coastal flooding, three mechanisms of coastal change were combined to evaluate cross-shore coastal retreat (after [54]):

$$\text{Coastal Erosion} = (\text{CCR}_{\text{SB}} + \text{CCR}_{\text{Climate}}) \times \text{T} + \text{CC}_{\text{Event}} \qquad (2)$$

where $\text{CCR}_{\text{SB}}$ is the pro-rated long-term (interannual- to decadal-scale) shoreline change rate [40], $\text{CCR}_{\text{climate}}$ is the coastal change rate associated with climate-change-induced factors (i.e., SLR) computed using the Bruun Rule [55], T is time in years, and $\text{CC}_{\text{Event}}$ is the event-based retreat, or retreat due to storms. To capture event-based erosion, a modification of the foredune erosion model presented by [56] was implemented [57] which assumes that the volume of sediment eroded from the foredune during a storm is deposited in the nearshore as the equilibrium beach profile shifts landward. This event-based erosion estimate is given as:

$$\text{CC}_{\text{Event}} = \frac{\text{T}_\text{D}}{\text{T}_\text{S}} \left( \frac{\left(\text{TWL}_{\text{maxyearly}} - \text{MHW}\right)\left(x_\text{b} - \frac{h_\text{b}}{\tan \beta_\text{f}}\right)}{\text{dhigh} - \text{MHW} + h_\text{b} - (\text{TWL}_{\text{maxyearly}} - \text{MHW})/2} \right) \qquad (3)$$

where $\text{T}_\text{D}$ is the storm duration, $\text{T}_\text{S}$ is the erosion response time scale, $\text{TWL}_{\text{maxyearly}}$ is the maximum yearly TWL, $x_\text{b}$ is the surf zone width from the mean high water (MHW) position determined using an equilibrium profile, $h_\text{b}$ is the water depth of wave breaking relative to MHW, $\tan \beta_\text{f}$ is the beach slope, and dhigh is the crest of the dune (extracted from lidar (light detection and ranging) data, 57).

This suite of coastal change models does not account for failure of BPS (due to a lack of enough information about the process) or retreat of bluff-backed beaches (<5% of the Tillamook County coast). On beaches backed by BPS, the beach was assumed to narrow at the rate of the total local chronic erosion ($\text{CCR}_{\text{SB}}$), resulting in dynamic beach slopes through the simulation period. Modeled beaches were further narrowed in the process of maintaining (i.e., raising to accommodate higher TWLs) and constructing BPS structures at a 2:1 slope. On nonhardened dune-backed beaches, beach slope was static as the dune erodes landward and equilibrium is reached.

The coastal flooding and erosion hazard models were intentionally relatively simple as the approach was designed to be modular and allow for the utilization of more sophisticated models (e.g., XBeach, [58]) when warranted.

### 2.1.2. Simulating Community Growth and Development

In addition to modeling physical landscape processes, human population growth and associated development processes were simulated using two submodels within *Envision*. The first submodel, *Target* (*Envision* Developers Manual, 2015), was used to grow and allocate population based upon a growth rate and a build-out capacity. The build-out

capacity was estimated prior to model simulation using zoning class and existing population distribution patterns. New population was spatially allocated proportionally to the difference between the existing density and the capacity surface, biased with preference factors reflecting circa 2010 population patterns (i.e., a preference to locate near the coastline or within a growth boundary). New development was allocated to the landscape based on population growth and the number of people per building in a separate *Envision* submodel, *Developer* (*Envision* Developers Manual, 2015).

*2.2. Evaluating Uncertainty*

Variability and uncertainty with respect to both climate change and human decision-making were expressed through policy and climate drivers. The derivation of those drivers and their use within the context of scenarios is described below.

### 2.2.1. Capturing Climate Uncertainty through Probabilistic Simulation of TWLs

The impact of climate change was analyzed through the perturbation of three individual climate drivers (Figure 3). Landscape metric uncertainty and variability was measured through 33 simulations of the 90-year period of 2010–2099 with high and low variations of SLR, wave climate, and the probability of occurrence of major El Niño events. Projections accounting for regional steric and ocean dynamics, cryosphere and fingerprinting effects, and vertical land motion, [24] were used to bound SLR projections. Possible climate-change-induced changes in the wave climate were based on significant wave height (SWH) distributions developed from the variability of statistically and dynamically downscaled projected global climate model estimates for the northeast Pacific Ocean [29,31,32]. Finally, the frequency of major El Niño events was varied between half of present and double present frequency [30]. In each simulation of an individual climate driver, current landscape conditions were maintained with no application of policies and the allocation of population onto the landscape was unrestricted by growth boundaries, thus highlighting the impact of each climatic driver. Thirty-three synthetic time series of current, or hereinafter baseline (i.e., no changes to sea level, wave climate, or El Niño frequency), climatic conditions were also simulated as a reference case.

### 2.2.2. Capturing Human Decision-Making Uncertainty through Stakeholder Derived Policy Options

Using *Envision*'s policy framework, human decision-making was represented across the landscape through an array of policies reflecting land management alternatives. The suite of six coproduced policies characterize reasonable actions that might be taken to build community adaptive capacity to climate change (Table 1). Each policy was first implemented individually under the 33 synthetic baseline climate simulations so that only changes resulting from that management decision were reflected in the results.

### 2.2.3. Capturing Uncertainty within the Context of Integrated Scenarios

In addition to simulating individual climate and policy drivers across the landscape, integrated climate and policy scenarios were used to evaluate and examine uncertainty and variability derived from the combination of physical and human drivers. Scenarios allow for exploration of feedbacks that may or may not be obvious when simulating only individual drivers (e.g., flooding may be exacerbated by the combination of both increasing wave heights and SLR). High-, medium-, and low-impact climate scenarios were derived around three SLR curves [24]. Within each scenario, the significant wave height and the frequency of major El Niño events could vary continuously within the bounds used during the simulation of the individual drivers (i.e., the frequency of major El Niño events varied between half and double the historic frequency). Combinations of three SLR scenarios, wave climate variability, and ENSO frequency projections were used to capture the inherent variability of the physical drivers through 33 probabilistic TWL simulations for each high-, medium-, and low-impact climate scenario, for a total of 99 scenarios overall (Figure 3).

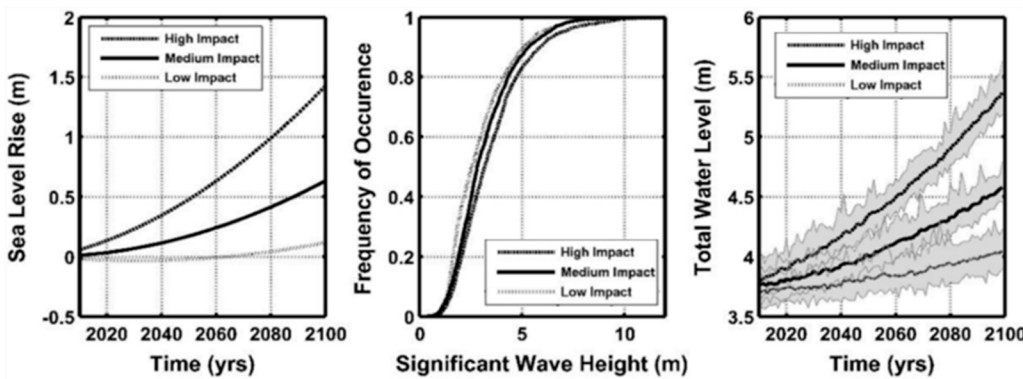

**Figure 3.** High-, medium-, and low-impact climate scenarios in terms of sea-level rise (SLR) from [20] (**left**), shift in wave climate from early to late century (**center**), and the mean yearly total water level (TWL) (**right**). The solid line in the significant wave height (SWH) probability distribution figure (**center**) represents a "present-day" SWH distribution. The dotted line to the right represents an increase to the present-day SWH distribution, while the dotted line to the left represents a decrease in the present-day SWH distribution by 2100. Bounds around the TWL (**right**) represent the max and min of the yearly average (after [43]).

**Table 1.** Policies implemented in the analysis of individual management drivers.

| Policy | Abbreviation | Description |
|---|---|---|
| 1 | BPS | Maintain current BPS and allow more BPS to be built on eligible lots. |
| 2 | Nourishment | Add beach nourishment to locations where beach access in front of BPS has been lost. |
| 3 | Easements | Remove buildings repetitively impacted by coastal hazards from within the hazard zone and establish conservation easements (e.g., managed retreat). |
| 4 | Relocate | Require movement of buildings frequently impacted by coastal hazards to a location above the Federal Emergency Management Agency's (FEMA) Base Flood Elevation (BFE) plus an additional 3ft and in the safest site of each respective lot. |
| 5 | Safest-Site | Construct new buildings above the Federal Emergency Management Agency's (FEMA) Base Flood Elevation (BFE) plus an additional 3 ft and in the safest site of each respective lot. |
| 6 | Hazard Zone | Determine Urban/Community Growth Boundaries (U/CGB) in accordance with the present-day policy but with prevention of new development within existing coastal hazard zones. |

In addition to climate scenarios, sets of individual policies were used to create four distinct policy scenario narratives (Table 2). In most cases, general policies were developed with variations specific to each policy scenario narrative. Each policy scenario was also simulated across all 99 integrated climate scenarios.

**Table 2.** Four policy scenario narratives iteratively codeveloped with local stakeholders. Each policy scenario contains a unique grouping of individual policies like those listed in Table 1.

| Policy Scenario | Scenario Narrative |
|---|---|
| Status Quo | Continuation of present-day policies. |
| Hold the Line | Policies or decisions were implemented that involve *resisting* environmental change in order to preserve existing infrastructure and human activities |
| Realign | Policies or decisions were implemented that involve *shifting development* to suit the changing environment (e.g., managed retreat). |
| Laissez-Faire | Current policies (state and county) were *relaxed* such that existing buildings, infrastructure and new development all trump the protection of coastal resources, public rights, recreational use, beach access, scenic views. |

## 3. Results and Discussion

Variation from baseline conditions and between scenarios, as well as uncertainty resulting from each of the climate and policy drivers, was quantified with respect to stakeholder defined landscape performance metrics including two metrics associated with exposure to coastal hazards, as expressed by 1) buildings impacted by flooding and 2) buildings impacted by erosion, and one metric related to public good as expressed by 3) beach accessibility. The two exposure metrics count the number of buildings impacted by flooding and erosion before being removed from the landscape. The beach accessibility metric relates to the percentage of time the beach is walkable and was defined as the ability to walk the beach at least 90% of the year during the maximum daily TWL. In addition to directly comparing the landscape performance metrics under baseline and variable driver conditions, the percent difference from the baseline value was calculated for decadal averages between 2030–2040, 2060–2070 and 2090–2100.

### 3.1. How Do Physical and Human Drivers Alter the Landscape through Time?

The variability with respect to landscape performance metrics was quantified for each of the six policies (Table 1) under a baseline climate scenario characterized by no changes to El Niño frequency, wave climate, sea level, or management of the landscape.

### 3.1.1. Impact of Individual Drivers

The difference in the landscape performance metrics resulting from the perturbation of the human or climate driver relative to the baseline scenario is used to evaluate the variability with respect to individual climate and human drivers. Figures 4, 6 and 8 display metric evolution through time under both the mean baseline and perturbed climate driver, while Figures 5, 7 and 9 display metric evolution through time under both the mean baseline and individual policy driver.

Figure 4 shows the number of buildings impacted by flooding for low- and high-SLR scenarios (a, b), low and high increases to wave heights (c, d), and halving or doubling the frequency of major El Niño events (e, f). High SLR was the most influential of the physical drivers in terms of impacts to buildings by flooding (Figure 4b). While the frequency of major El Niño events did not significantly shift metric values from the baseline (Figure 4e,f), changes to the wave climate did have an influence on flooding (Figure 4c,d), albeit relatively minor, particularly in the latter half of the century. Even under the baseline scenario there was an increase in the average number of buildings impacted through time (Figure 4), primarily due to continued development near the coast.

In contrast, Figure 5 shows the number of buildings impacted by flooding under the policy drivers (a) Policy 1: BPS, (b) Policy 2: Nourishment, (c) Policy 3: Easements, (d) Policy 4: Relocate, (e) Policy 5: Safest-Site, and (f) Policy 6: Hazard Zones. The number of building impacted by flooding was sensitive to the construction of BPS, the nourishment of beaches fronting BPS, the formation of easements to facilitate managed retreat, and the relocation of buildings to safer areas within a parcel (Figure 4a–d). Overall, individual policy drivers had a larger variation between policies on the number of buildings impacted by flooding, relative to the baseline, than individual climate drivers (Figure 5).

While installing BPS protected property from erosion, those protected properties ultimately experienced greater levels of flooding due to the modification of local morphology. Because BPS prevents landward migration of the backshore, the long-term erosion rate due to sediment budget factors caused the beach to narrow and steepen, thus increasing wave runup and eventually extreme TWLs. BPS were maintained through time to prevent overtopping; however, the height of BPS was limited to preserve current view sheds. Additionally, raising the elevation of the BPS structure crest forced the extension of the structure horizontally, further narrowing the beach, a feedback which resulted in increased exposure to coastal flooding.

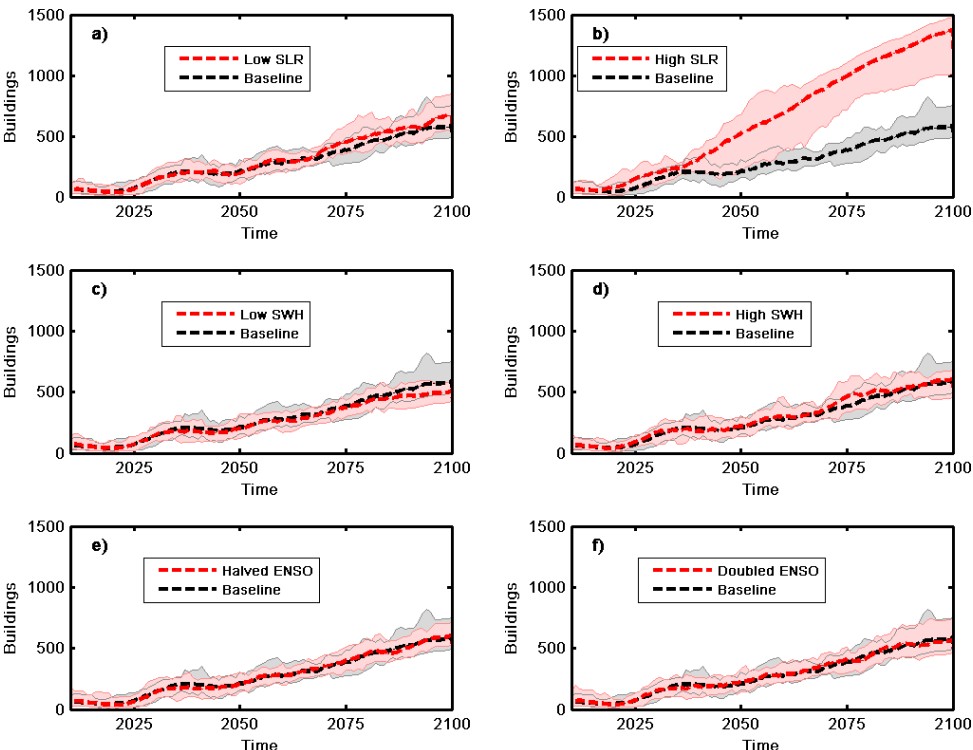

**Figure 4.** Number of buildings impacted by flooding through time under each of the six climate drivers (panels **a–f**) compared to the baseline. Dashed lines indicate the decadal mean of all simulations. Bounds represent the minimum and maximum decadal means.

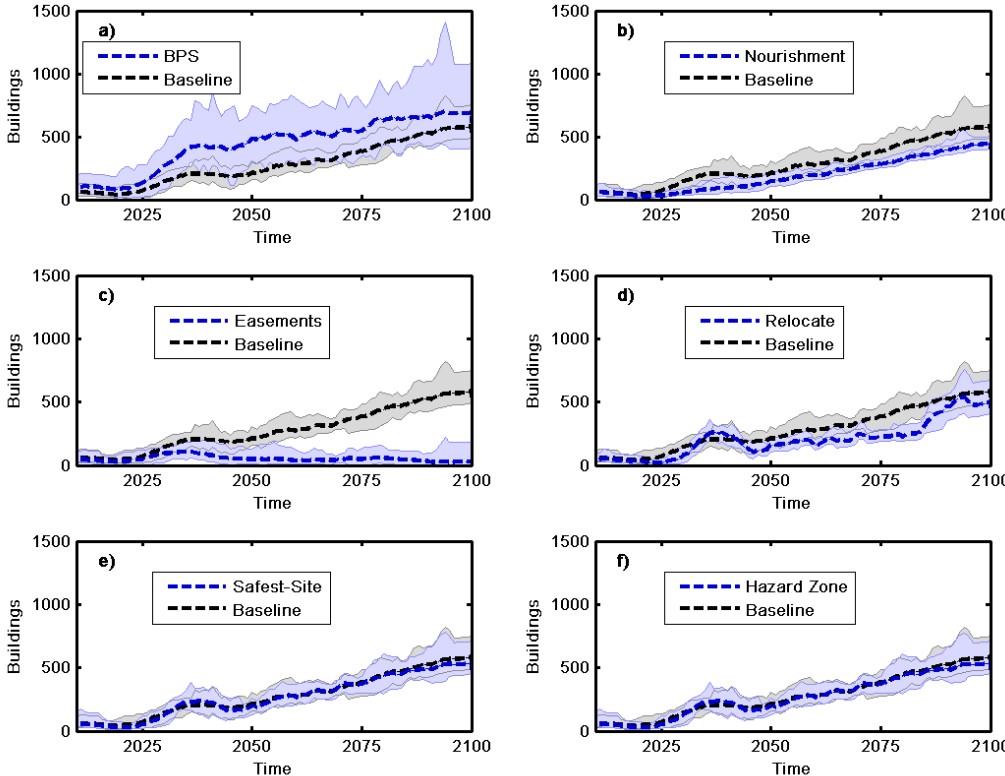

**Figure 5.** Number of buildings impacted by flooding through time under each of the six policy drivers (panels **a–f**) compared to the baseline. Dashed lines indicate the decadal mean of all simulations. Bounds represent the minimum and maximum decadal means.

The addition of sediment onto the beach through the process of beach nourishment reduced flooding impacts through the widening and flattening of the beach and subsequent reduction in TWLs (Figure 5b, Policy 2). Simulated easements effectively move buildings outside of the hazard zone, whereas relocating buildings to the safest site within the existing parcel was only partially effective (Figure 5c,d, Policies 3 and 4). Model results indicate that restrictions on new development (i.e., hazard zone implementation and the requirement to construct new homes only in the safest site of a parcel) reduce impacts only minimally by the end of the century—primarily due to the low projected growth rate within the county (0.39%–0.78% per year, [59], Figure 5e,f, Policies 5 and 6). Higher levels of deviation from the baseline would be expected if a greater population, and thus more development, was projected for the region in the future.

Figure 6 displays the second exposure metric, the number of buildings impacted by erosion. The number of buildings impacted by erosion deviated from the baseline the most under the heightened SLR (Figure 6). The increasing trend under all simulations was a result of buildings impacted by (1) the background shoreline change rate related to sediment budget factors, which is applicable under all climate drivers, (2) the shoreline retreat due to SLR, and/or (3) increased erosion during storm events. Thus, near the end of the century, the model indicates almost 600 structures would be impacted by the greatest yearly TWL event under the baseline climate scenario. The introduction of a sediment budget factor into the model based upon historical trends reduces the variability with respect to climate in the number of buildings impacted by erosion. This long-term signal may obstruct potential sensitivity to both ENSO frequency and shifts in SWH.

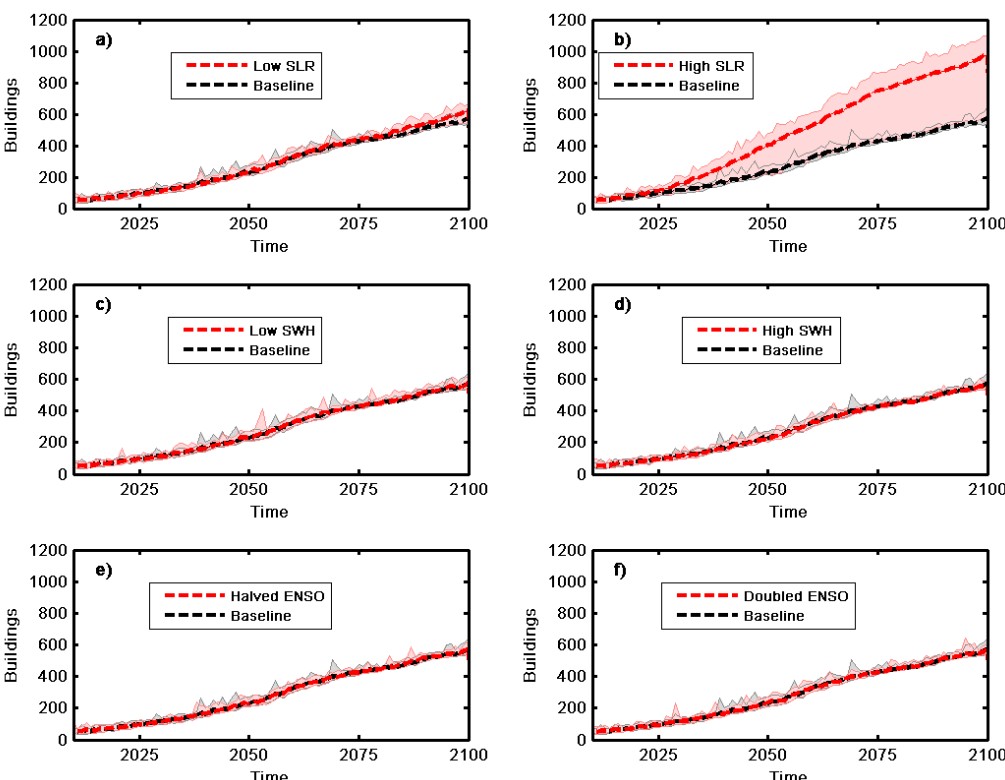

**Figure 6.** Number of buildings impacted by erosion through time under each of the six climate drivers (panels **a**–**f**) compared to the baseline. Dashed lines indicate the decadal mean of all simulations. Bounds represent the minimum and maximum decadal means.

Figure 7 shows the number of buildings impacted by erosion under the individual policy drivers. Three of the modeled individual policy drivers had an impact on the number of buildings impacted by erosion; the construction of BPS (Policy 1, Table 1), the formation of easements in response to hazard exposure (Policy 3), and the relocation of

existing buildings to the safest site within a parcel (Policy 4, Figure 7). Both the creation of easements and the construction of new BPS essentially eliminated exposure to erosion hazards (Figure 7a,c) whereas the relocation of buildings only reduced the exposure by approximately 100 buildings by the end of the century (Figure 7d). The easement policy (Policy 3) completely removed structures from areas impacted by hazards while the relocate policy (Policy 4) only delayed the ultimate exposure to erosion by simply moving existing structures within an existing parcel. Beach nourishment had no impact on the number of buildings impacted by erosion as the policy was applied only at locations where BPS had been previously constructed (Figure 7b).

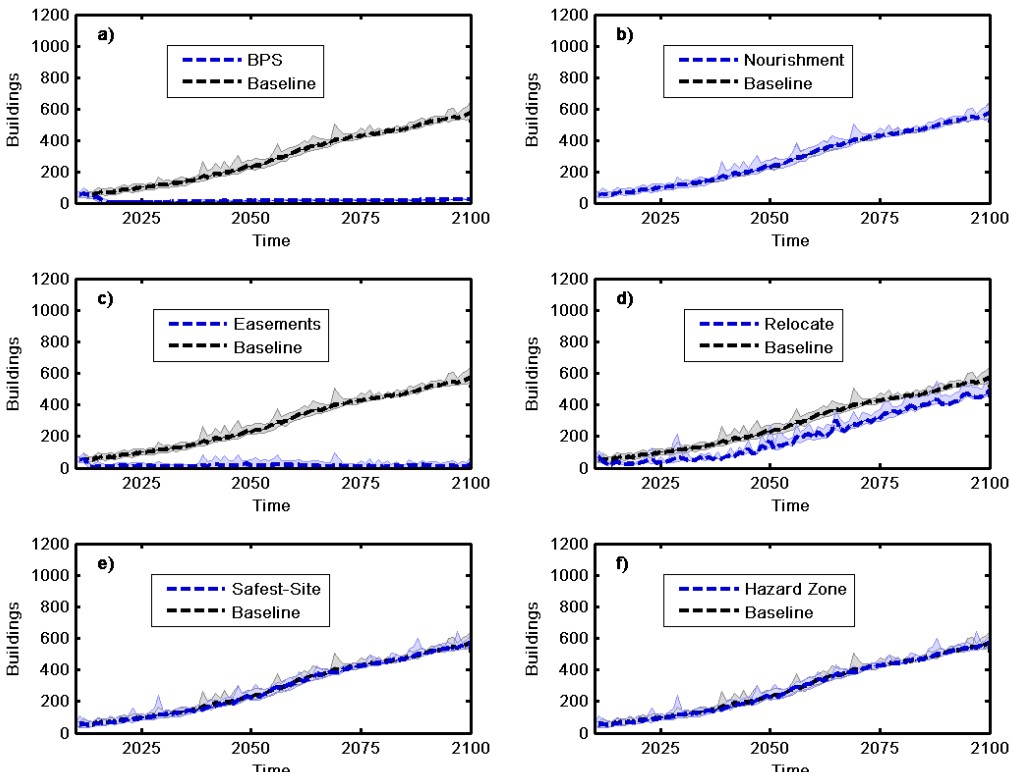

**Figure 7.** Number of buildings impacted by erosion through time under each of the six policy drivers (panels **a–f**) compared to the baseline. Dashed lines indicate the decadal mean of all simulations. Bounds represent the minimum and maximum decadal means.

Figure 8 shows the third metric, beach accessibility under each climate driver. Beach accessibility, was also impacted by both SLR and wave climate (Figure 8). High SLR alone decreased beach accessibility from approximately 80% of the Tillamook County coastline to less than 60% (Figure 8b). Should Tillamook county experience lower significant wave heights into the future, results indicate that accessibility would increase by up to 10% relative to the baseline (Figure 8c). Raising wave heights by the same margin had less of an impact, decreasing accessibility by less than 5% (Figure 8d). As with the two performance metrics related to coastal hazard exposure, variability in the frequency of major El Niño events produced only minimal variations from the baseline (Figure 8e,f).

Figure 9 displays beach accessibility under each individual policy driver. Model results suggest that individual policy drivers had less of an impact on beach accessibility than climate drivers (Figure 9). The most significant variation from baseline occurred with BPS construction, which reduced accessibility by less than 10% (Figure 9a). Including a nourishment policy kept accessibility essentially constant through the century (Figure 9b). Other modeled policies had no effect on beach accessibility.

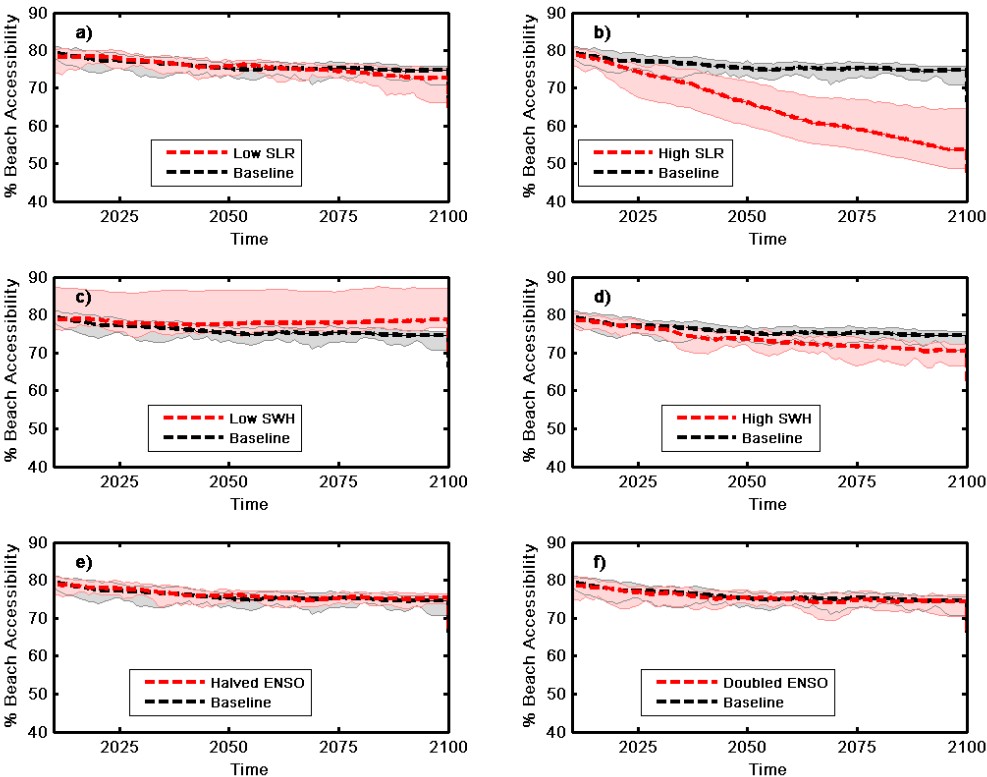

**Figure 8.** Beach accessibility through time under each of the climate drivers (panels **a**–**f**). Dashed lines indicate the decadal mean of all simulations. Bounds represent the minimum and maximum decadal means.

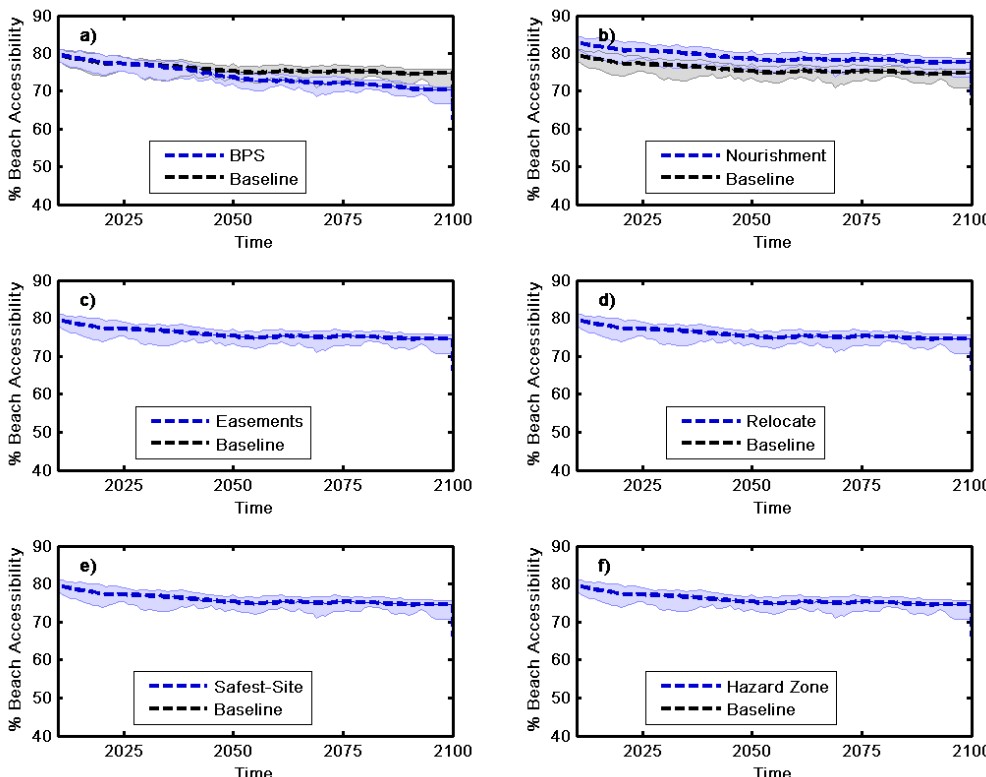

**Figure 9.** Beach accessibility through time under each of the six policy drivers (panels **a**–**f**). Dashed lines indicate the decadal mean of all simulations. Bounds represent the minimum and maximum decadal means.

Modeled deviations from the baseline were asymmetric and did not linearly correspond to the perturbation in the driver variable. Figure 10 reveals the percent difference in each of the landscape performance metric values between each of the climate and policy drivers and the baseline scenario over three time periods representing early century (2030–2040), midcentury (2060–2070), and late century (2090–2100). Error bars indicate early in the century, human adaptation strategies in the form of easement creation, BPS construction, and beach nourishment overwhelmed climate drivers in two of the three performance metrics analyzed in terms of variation from the baseline (Figure 10). While construction of BPS resulted in a reduction of the number of buildings impacted by erosion, it increased exposure to flooding and reduced beach accessibility. In contrast, nourishing the beach fronting BPS reduced wave runup, thus increasing beach access and reducing overtopping of the structure crest. Easements reduced coastal hazards exposure by almost 100% and thus had the greatest benefit of any policy. Policies that modified future new development patterns had less of an impact on the landscape due to Tillamook County's low projected population growth rate.

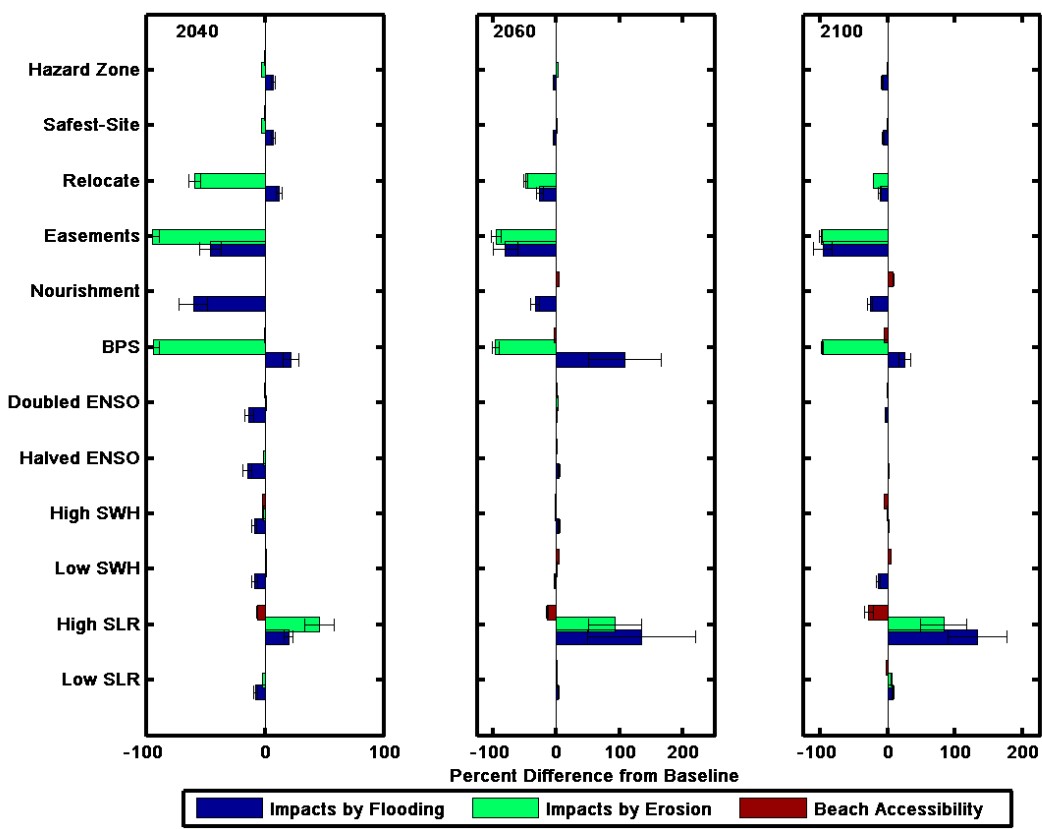

**Figure 10.** The percent difference in each of the landscape performance metric values between each of the climate and policy drivers and the baseline scenario. Values are presented for three time periods, 2030–2040 (**left**), 2060–2070 (**middle**), and 2090–2100 (**right**). Error bars indicate the standard deviation in the percent difference from the baseline.

SLR was the physical climate driver with the greatest impact on the three landscape performance metrics. The county-wide variation in beach accessibility was minimal when compared to the number of properties impacted by coastal hazards and was only influenced significantly by high SLR. By midcentury, the greatest variation from a baseline climate was under a high-SLR climate driver, which by 2100 increased the impact of flooding and erosion, and decreased beach accessibility by approximately 150%, 75%, and 30%, respectively.

### 3.1.2. Impact of Drivers Integrated as Scenarios

The three landscape performance metrics described above were also compared under combined climate impact and policy scenarios (Figure 11). Here, variability associated with climate was computed as the range of the mean high- and low-climate-impact climate scenarios within any of the four policy scenarios (shading limits in Figure 11). The range associated with each policy scenario is the difference between the means of the policy scenarios (dashed colored lines in Figure 11).

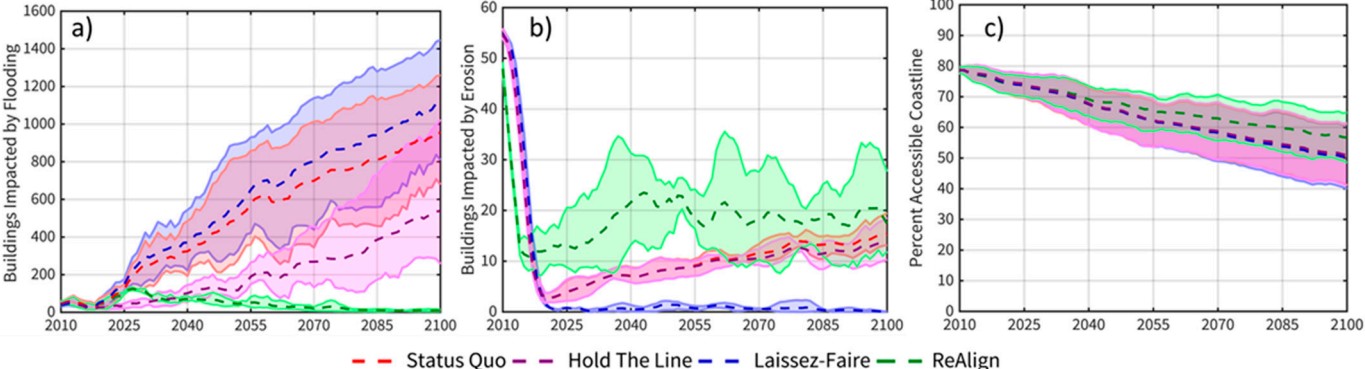

**Figure 11.** Three landscape performance metrics (buildings impacted by flooding, (**a**); buildings impacted by erosion, (**b**); beach accessibility, (**c**) compared across four policy scenarios under a range of climate scenarios. Dashed lines indicate the mean of the medium-impact climate scenarios. Shading indicates the mean of the low- and high-impact climate scenarios.

Greater variability in metric response was observed within policy scenarios than individual policy drivers as each scenario contains multiple policies and was simulated under 99 climate scenarios with varying shifts of ENSO frequency and SWH (grouped by low, medium, and high SLR). The increase in the number of buildings impacted by flooding during the first half of the century was in response to both climate drivers and policy drivers, predominately in the form of SLR and the construction of BPS (Figure 11a). A feedback not observed through our analysis of only individual drivers was the increased levels of BPS construction in response to high SLR. The greatest impacts to buildings by flooding occurred under the *Laissez-Faire* policy scenario, both because BPS construction was permitted without restriction and because the preference for new development near the shoreline was increased. Flooding impacts under the *Realign* policy scenario were reduced by the end of the century due to both the relocation of people and development away from coastal hazard zones and the limitation of further BPS construction.

The relative magnitude of variation between both policy scenarios and climate scenarios was significantly less for buildings exposed to erosion hazards (Figure 11b). Erosional trends were much different from the baseline trend observed in the previous section due to both the construction of BPS (in the *Status Quo*, *Laissez-Faire*, and *Hold the Line* policy scenarios) and the formation of easements under the *Realign* policy scenario. These two management options reduced the magnitude of erosion impacts by almost two orders of magnitude, even in a high-SLR scenario. Thus, in all scenarios, erosion had far less of an impact on the landscape than flooding. Near the end of the century, the impacts of climate and sediment budget factors began to overtake properties not eligible (under current Oregon state law) for BPS construction in the *Status Quo* and *Hold the Line* policy scenarios. The *Laissez-Faire* policy scenario had the fewest number of buildings impacted by erosion as property owners constructed BPS regardless of current eligibility status. The lack of BPS construction in the *Realign* policy scenario resulted in greater impacts to buildings by erosion and greater variability with respect to climate scenarios.

By 2100, the combination of climate impacts and hardening of the shoreline significantly reduced beach accessibility across all scenarios (Figure 11c). Greater accessibility was maintained under the *Realign* and *Hold the Line* policy scenarios and reduced under

the *Status Quo* and *Laissez-Faire* scenarios. Beach nourishment in the *Hold the Line* scenario did not maintain beach accessibility under all climate impact scenarios and was ineffective under the medium- and high-impact climate scenarios as the BPS were extended onto the beach in response to higher TWLs.

The relative influence of climate and policy varied when considered as individual drivers and within the context of scenarios (Appendix A). Generally, the consequences of both climate drivers and human adaptations were exacerbated through time under all metrics both within the context of scenarios and as individual drivers. Both policy and climate had significant impacts on the three performance metrics evaluated. When compared between policy scenarios, the number of buildings impacted by flooding were more variable than when compared between climate scenarios. The number of buildings impacted by erosion also varied with respect to human adaptation decisions both within scenarios and when individual drivers were evaluated. In contrast, climate had the largest influence on beach accessibility within the context of policy scenarios and as individual drivers.

Comparing individual drivers to a baseline allows for exploration of metric sensitivity to specific perturbations, whereas comparing metric results across scenarios allows for comparison of potential feedbacks. For example, the impact of BPS across the landscape was much greater in a high-SLR scenario, not only because a higher percentage of the shoreline was armored in response to more frequent erosion, but also because increased armoring changed the coastal morphology, thus exacerbating flooding later in the century.

### 3.2. How Do Climate and Policy Drivers Change Landscape Performance Metric Uncertainty over Time?

Quantifying the uncertainty through time under each driver and scenario is important in providing robust assessment of adaptation strategies and management options under a range of climate scenarios. While there are many forms of uncertainty within the modeling process described here, uncertainty as expressed in this analysis is equal to the spread in landscape performance metrics resulting from probabilistic simulations of daily maximum TWLs. The relative coefficient of variance, or the ratio of the standard deviation to the mean, provides a measurement of uncertainty within the 33 simulations under each climate and policy driver.

#### 3.2.1. Impact of Individual Drivers on Metric Uncertainty

Variance over time is measured for each individual driver and compared to the baseline in Figure 12. The highest levels of variance were found in the number of buildings impacted by flooding (Figure 12a,d). Under all climate drivers, uncertainty with respect to flooding decreased through time, particularly under high SLR. The overall increase in magnitude and decrease in uncertainty was likely due to the presence of BPS and an increase in yearly maximum TWLs through time. Within the model, once BPS has been constructed to its maximum level and the beach had been narrowed such that there was frequent overtopping, the same locations were likely to be impacted by flooding on a regular basis. At some point, a threshold was reached under which the same properties experienced flooding consistently during the maximum TWL event of the year. Surpassing this threshold was accelerated and exacerbated under the high-SLR scenario. There was more uncertainty across the policy drivers than climate drivers with respect to flooding impacts. Easements produced a higher relative coefficient of variance throughout the century because buildings that were regularly flooded were removed from the hazard zone, thus the remaining buildings were impacted with greater temporal irregularity. Finally, the construction of new BPS indicated higher uncertainty in the mid- to late century in metrics related to flooding.

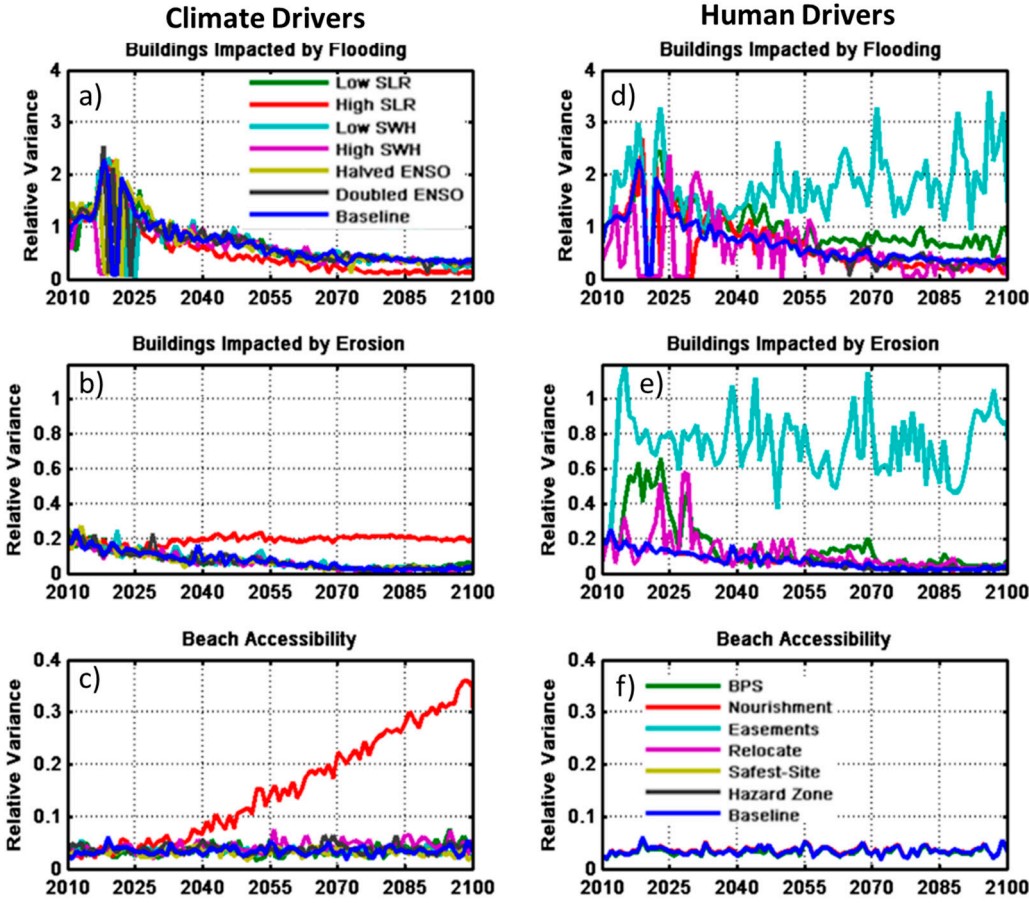

**Figure 12.** Relative coefficient of variance for each of the climate (left, **a–c**) and policy (right, **d–f**) drivers through time.

Similar to flooding hazards, variance within the number of buildings impacted by erosion decreased over time with the exception of under the high-SLR climate driver (Figure 12b). Again, there was higher uncertainty under the policy drivers than the climate drivers, particularly within the creation of easements, relocation of buildings, and construction of new BPS, but this uncertainty converged to the baseline levels after the first quarter century for the latter two drivers (Figure 12e).

Uncertainty in beach accessibility under all drivers, except for high SLR, remained fairly constant through time and was smaller than uncertainty for the number of buildings impacted by erosion or flooding (Figure 12c,f). This indicates that under all but a high-SLR scenario, the portion of beach within Tillamook County that remains accessible at least 90% of the year is constant, whereas under a high-SLR scenario, the accessibility of a segment of coastline from year to year is less predictable.

Figure 13 shows the percent difference in the coefficient of variance for each of the landscape performance metrics between each of the climate and policy drivers and the baseline scenario. The majority of the individual climate and policy drivers increased the coefficient of variance within the three metrics examined (Figure 13). Overall, policy drivers had a much greater impact on increasing uncertainty in impacts to buildings by coastal hazards throughout the century. However, no policy driver shifted uncertainty with respect to beach accessibility by greater than 10%. The construction of BPS and formation of easements generally increased uncertainty with respect to flooding and erosion hazards. Beach nourishment trends reversed near the end of the century, and variance with respect to buildings impacted by flooding was reduced. The relocation of buildings early in the century increased variance with respect to erosion but decreased variance with respect to flooding. Implementing hazard zones with respect to new development and enforcing safest site construction in coastal areas resulted in no change in uncertainty

until midcentury, at which time the two policies decrease uncertainty for metrics related to erosion and increase uncertainty for metrics related to flooding hazards.

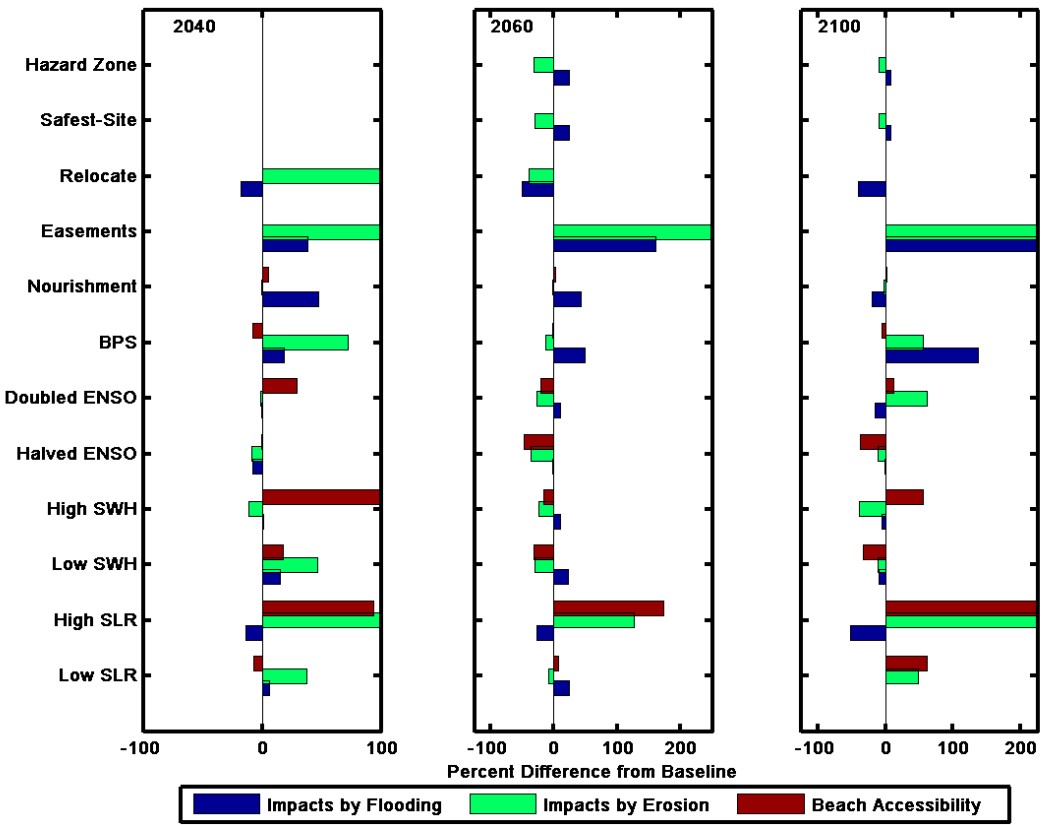

**Figure 13.** The percent difference in the coefficient of variance for each of the 3 metric values between each of the climate and policy drivers and the baseline scenario. Values are presented for three time periods, 2030–2040 (**left**), 2060–2070 (**middle**), and 2090–2100 (**right**).

Of the six climate drivers, only high SLR resulted in a persistent pattern of heightened uncertainty with respect to buildings impacted by erosion and beach accessibility (Figure 13). As mentioned above, within the model there was a decrease in variance due to the sustained inundation of coastal properties. Early in the century, the uncertainty in beach accessibility caused by a positive shift in SWH was greater than any other driver, physical or human. This trend was not maintained through the end of the century and was in fact reversed midcentury. Unsurprisingly, decreasing the frequency of ENSO events reduced variance with respect to beach accessibility, while doubling the frequency had mixed effects throughout the century. By the end of the century, increases to climate drivers increased uncertainty in beach accessibility, whereas decreases to these drivers (SWH, ENSO frequency) decreased variance.

3.2.2. Impact of Drivers Integrated as Scenarios on Metric Uncertainty

Uncertainty, quantified using the coefficient of variance, was also examined with respect to the integrated policy and climate scenarios (Figure 14). Uncertainty attributed to climate was calculated within each policy scenario across all 99 climate simulations (Figure 14a–c). Uncertainty attributed to human decisions was calculated under each climate impact scenario (high, medium, and low) across all four policy scenarios (Figure 14d–f). In three out of the four policy scenarios, uncertainty due to climate decreased over time for the metric of flooding impacts to buildings (Figure 14a). The exception occurred under the *Realign* scenario, which reflected the increased uncertainty driven by the formation of easements.

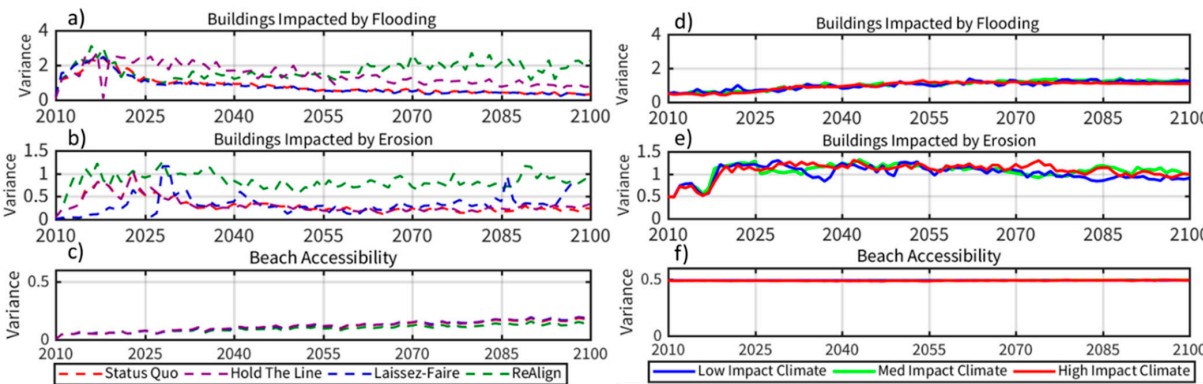

**Figure 14.** Coefficient of variance with respect to climate under each policy scenario (left, **a**–**c**) and with respect to policy scenarios under a high-, medium-, and low-impact climate scenario (right, **d**–**f**).

Similar to the buildings impacted by flooding, the number of buildings impacted by erosion was most uncertain with regards to climate under the *Realign* scenario (Figure 14b). Climate uncertainty under the *Laissez-Faire* policy scenario lacked a distinct trend. The number of impacted buildings in the *Laissez-Faire* policy scenario was minimal, thus causing sensitivity in the coefficient of variance to small differences within the climate simulations. Uncertainty in beach accessibility increased due to climate impacts under all policy scenarios, the most so in policy scenarios in which BPS were constructed (Figure 14c).

Under each climate scenario, relative trends and magnitudes of uncertainty between policy scenarios were essentially equal (Figure 14d–f). The coefficient of variance between policy scenarios increased over time with respect to buildings impacted by flooding and erosion (Figure 14d,e). There was no measured change in variance between policy scenarios under any climate scenario in reference to beach accessibility (Figure 14f).

Comparing the variance within the context of scenarios indicates that the uncertainty in landscape performance metrics with respect to climate is dependent upon human decisions, whereas policy scenario uncertainty was generally consistent across a range of climate drivers.

The maximum variance and general trends with respect to policy and climate uncertainty both within scenarios and as individual drivers is illustrated in Table 3. Trends within scenarios and in individual drivers were inconsistent across metrics. The metric with the highest uncertainty was buildings impacted by flooding. This uncertainty was greatest under individual management options and typically decreased through time. For erosion hazards, uncertainty was greater within scenarios. Uncertainty in buildings impacted by erosion due to individual human and physical drivers decreased through time, whereas uncertainty between policy scenarios increased. The minimal uncertainty of buildings impacted by erosion can be attributed to the shoreline change rate related to sediment budget factors. Beach accessibility was less uncertain under almost all scenarios and drivers.

**Table 3.** Magnitude and trend in relative variance (uncertainty) through time.

| | | Buildings Impacted by Flooding | | Buildings Impacted by Erosion | | Beach Accessibility | |
|---|---|---|---|---|---|---|---|
| | | Max. Rel. Variance | General Trend | Max. Rel. Variance | Trend | Max. Rel. Variance | Trend |
| **Individual Drivers** | Climate Uncertainty | 2.6 | Decrease | 0.2 | Decrease | 0.4 | Increase |
| | Policy Uncertainty | 3.7 | Decrease | 1.1 | Decrease | <0.1 | No Change |
| **Within Scenarios** | Climate Uncertainty | 3.1 | Decrease | 1.3 | Static | 0.2 | Increase |
| | Policy Uncertainty | 1.2 | Increase | 1.3 | Increase | 0.5 | No Change |

## 4. Conclusions

Globally, coastal communities are increasingly faced with the impacts of climate change. The combination of sea level rise, changes to patterns of storminess, and evolving development pressures has the potential to significantly increase the effects of flooding and erosion on coastal populations. The strategies used to adapt to these impacts have the potential to either improve or exacerbate exposure to hazards. Understanding the impact that each decision has on the landscape in combination with how that policy driver influences the uncertainty of future projections of hazard impacts can more robustly inform decisions. Using the spatially explicit, policy-centric modeling platform, *Envision*, the relative impact to stakeholder relevant exposure metrics of six climate drivers and six policy drivers (related to management options) was quantified with respect to baseline conditions. Impacts were also quantified under a set of integrated climate scenarios grouped by SLR and policy scenarios of grouped management options to allow a more thorough exploration of feedbacks between climate and policy.

Variability and uncertainty were measured across three coproduced landscape performance metrics including (1) number of buildings impacted by flooding, (2) number of buildings impacted by erosion, and (3) beach accessibility in order to capture impacts of climate change and adaptation measures to both the built and natural environments. Variability in beach accessibility was greatest due to climate drivers within scenarios; for buildings impacted by erosion, variability was greatest with respect to individual policies; and for buildings impacted by flooding, variability was greatest between policy scenarios. In general, variability with respect to both climate and policy increased over time. Trends in uncertainty decreased, remained static, or increased through time depending upon the metric and driver. Uncertainty was greatest for the metric of flooding hazards and least for beach accessibility. Uncertainty in all landscape metrics with respect to climate was dependent upon policy decisions, whereas uncertainty associated with policy decisions was generally consistent across a range of climate drivers.

Overall, while adaptation policies produced a greater deviation from baseline conditions, climate change produced the greatest variance through time. Based upon the assumptions used in this modelling effort, the policies implemented in response to coastal hazards have a greater impact on community exposure than climate change. Quantifying variability and uncertainty within the *Envision* framework helped improve the relevance of model results to the project's stakeholders (Tillamook County Knowledge to Action Network) by allowing for the determination of the relative impact of policy and management decisions on the adaptive capacity of Pacific Northwest coastal communities under a range of future climate scenarios. Implementation of policies for which the outcome is less certain under the full range of climate scenarios may be less desirable than the implementation of a policy simulated to positively impact the metric under all climate scenarios. While no alternative presented here is a specific forecast representing the future of Tillamook County, the range of results presented is allowing stakeholders to constrain the deep uncertainty associated with their climate change adaptation planning. Understanding the impacts of decisions and climate both as individual drivers and coupled within scenarios can potentially allow for more robust and informed determination of best practices with respect to various adaptation pathways, within the constraints of the modeled representation of coastal community drivers and processes.

**Author Contributions:** Conceptualization, P.R., J.P.B., K.A.S., A.K.M., E.L.; methodology, A.K.M., P.R., J.P.B., K.A.S., and E.L.; software, J.P.B., K.A.S., A.K.M., P.R.; validation, A.K.M., P.R., J.P.B., K.A.S., and E.L.; formal analysis, A.K.M., K.A.S.; investigation, A.K.M., P.R., J.P.B., K.A.S., and E.L.; resources, J.P.B.; data curation, K.A.S., A.K.M.; writing—original draft preparation, A.K.M.; writing—review and editing, A.K.M., P.R., J.P.B., K.A.S., and E.L.; visualization, A.K.M., K.A.S., and P.B.; supervision, P.R. and J.P.B.; project administration, P.R.; funding acquisition, P.R. and J.P.B. All authors have read and agreed to the published version of the manuscript.

**Funding:** This study was funded by the National Oceanic and Atmospheric Administration's (NOAA) Coastal and Ocean Climate Applications (COCA) program under NOAA grants NA12OAR4 310109 and NA12OAR4310195 and NOAA's Regional Integrated Sciences and Assessments Program (RISA), under NOAA grant NA10OAR4310218 and NA15OAR4310145. Additionally, we thank the members of the Tillamook County Coastal Futures Knowledge-to-Action-Network for their hard work and dedication. PR and JB were supported by Oregon Sea Grant grant # during the final stages of manuscript preparation.

**Data Availability Statement:** Publicly available datasets were analyzed in this study. This data can be found here: https://tidesandcurrents.noaa.gov/ (accessed on 24 February 2021).

**Acknowledgments:** Tide gauge records used to initialize the TWL-FSM model are available through the National Oceanic and Atmospheric Administration (NOAA) National Ocean Service (NOS) website. We thank Melisa Menendez and Jorge Perez at the Environmental Hydraulics Institute of the Universidad de Cantabria (IH Cantabria) for providing Global Ocean Wave 2 (GOW2) data which was also used in initializing the TWL-FSM model.

**Conflicts of Interest:** The authors declare no conflict of interest.

## Appendix A

**Table A1.** Maximum ranges of three landscape performance metric values associated with climate and human drivers. As individual drivers, the range is measured by the maximum absolute difference from the baseline value in any year during the 90-year time series. Within scenarios, the range associated with climate is computed as the maximum range of the high and low climate impact means within any of the four policy scenarios and the range associated with policy is the greatest difference between the mean of the maximum metric value of all policy scenarios and minimum metric value of all policy scenarios.

| | Variability | Buildings Impacted by Flooding | Buildings Impacted by Erosion | Beach Accessibility |
|---|---|---|---|---|
| **As Individual Drivers** | Max. Range Associated with Climate | 840 Buildings | 411 Buildings | 33% |
| | Max. Range Associated with Human Decisions | 610 Buildings | 555 Buildings | 23% |
| **Within Scenarios** | Max. Range Associated with Climate | 1780 Buildings | 174 Buildings | 35% |
| | Max. Range Associated with Human Decisions | 1922 Buildings | 178 Buildings | 24% |

## Appendix B

**Table A2.** List of symbols and acronyms.

| Acronym or Symbol | Definition |
| --- | --- |
| **BPS** | Backshore protection structure |
| **CC$_{event}$** | Event-based erosion |
| **CCR$_{climate}$** | Coastal change rate associated with climate change-induced factors (i.e., SLR) computed using the Bruun Rule |
| **CCR$_{SB}$** | Pro-rated long-term (interannual- to decadal-scale) shoreline change rate |
| **dhigh** | Dune crest height |
| **ENSO** | El Niño Southern Oscillation |
| **h$_b$** | Water depth of wave breaking relative to MHW |
| **KTAN** | Knowledge to action network |
| **MHW** | Mean high water |
| **MSL** | Mean sea level |
| **PNW** | Pacific Northwest |
| **R$_{2\%}$** | Two percent exceedance value of vertical wave runup on a beach or structure above the still water level |
| **SLR** | Sea level rise |
| **SWH** | Significant wave height |
| **T** | Time |
| **Tan β$_f$** | Beach slope |
| **T$_D$** | Storm duration |
| **T$_S$** | Erosion response time scale |
| **TWL** | Total water level |
| **x$_b$** | Surf zone width from MHW position determined using an equilibrium profile |
| **η$_A$** | Deterministic astronomical tide |
| **η$_{NTR}$** | Nontidal residual generated by physical processes including wind setup and barometric surge |

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
