# Peer review of "Quantifying Uncertainty in Exposure to Coastal Hazards Associated with Both Climate Change and Adaptation Strategies: A U.S. Pacific Northwest Alternative Coastal Futures Analysis"

_water, doi:10.3390/w13040545_

Round 1

Reviewer 1 Report

The authors proposed an interesting topic analyzing uncertainty in exposure to coastal hazards associated with both climate change and adaptation strategies.
The work was thoroughly written, the authors carried out detailed analyzes, the results are very
significant as well as the subject of the topic under consideration.
This paper is strongly recommended for publication.

Only few points need revision:
-Figures : please enlarge Figures 1 and 10 because they can hardly be read.

Author Response

We thank the reviewer for the thoughtful description of our paper.  Both Figures 1 and 10 have been modified to enhance readability.

Reviewer 2 Report

A BRIEF SUMMARY

The paper titled “Quantifying uncertainty in exposure to coastal hazards associated with both climate change and adaptation strategies: A U.S. Pacific Northwest alternative coastal futures analysis”, presents a good topic for readers of this Journal. However I suggest that the authors try to add some more references especially in the "part 1 (introduction)" of the paper. I have indicated some suggestions, but more can be added which would make the foundation for the arguments stronger.

SPECIFIC COMMENTS

I strongly suggest the following studies on risk analysis in coastal area.

  • Qu, K., Yao, W., Tang, H.S. et al. Extreme storm surges and waves and vulnerability of coastal bridges in New York City metropolitan region: an assessment based on Hurricane Sandy. Nat Hazards (2021). https://doi.org/10.1007/s11069-020-04420-y

  • Apollonio, C.; Bruno, M.F.; Iemmolo, G.; Molfetta, M.G.; Pellicani, R. Flood Risk Evaluation in Ungauged Coastal Areas: The Case Study of Ippocampo (Southern Italy). Water 2020, 12, 1466. https://doi.org/10.3390/w12051466

Author Response

We thank the reviewer for their comments. This reviewer suggests that we add additional citations to our manuscript. However, after careful review, it was determined that neither of these two suggested additional citations are relevant to the current manuscript. Further, we note that our Introduction section already cites 38 manuscripts, many of which focus on the topic of risk analysis in coastal areas.

Reviewer 3 Report

This paper provides a solid technical analysis of the impacts of climate change and adaptation strategies on flooding, erosion, and recreational beach accessibility. Discussion is focused on uncertainties associated with climate change versus uncertainties stemming from adaptation strategies.

Strengths:

  1. The paper marries the physical aspects of climate change with societal efforts to mitigate climate risks. Integration of policy science and climate science offers potential for new insights into hazard conditions in coastal Oregon.
  2. The writing is crystal clear as per the abstract, introduction, analysis, and conclusions.
  3. The paper reveals the very important finding that uncertainties associated with policy decisions introduce more uncertainty about flooding, erosion, and accessibility than climate change uncertainty. This is an important message for hazard managers and adaptation planners.

Weaknesses

  1. It was not clear to me how the authors of this paper engaged with regional stakeholders. They indicate that stakeholders provide input to scenario development, but their specific input and the processes of obtaining it and using it for scenario development is not clear. I would ask the authors to be more specific about stakeholder input, emphasizing what stakeholders contributed to this effort and what they will do with the products of this research.
  2. Modeling experiments emphasize technical aspects of measuring uncertainty. I would ask the authors to explore in the conclusions the implications of these uncertainties for hazard management. How can hazard management professionals make better adaptation decisions in the face of inevitable uncertainty about the climate and policy environment.

Author Response

This paper provides a solid technical analysis of the impacts of climate change and adaptation strategies on flooding, erosion, and recreational beach accessibility. Discussion is focused on uncertainties associated with climate change versus uncertainties stemming from adaptation strategies.

Strengths:

  1. The paper marries the physical aspects of climate change with societal efforts to mitigate climate risks. Integration of policy science and climate science offers potential for new insights into hazard conditions in coastal Oregon.
  2. The writing is crystal clear as per the abstract, introduction, analysis, and conclusions.
  3. The paper reveals the very important finding that uncertainties associated with policy decisions introduce more uncertainty about flooding, erosion, and accessibility than climate change uncertainty. This is an important message for hazard managers and adaptation planners.

 We thank the reviewer for the description of the key strengths of our paper.

Weaknesses

  1. It was not clear to me how the authors of this paper engaged with regional stakeholders. They indicate that stakeholders provide input to scenario development, but their specific input and the processes of obtaining it and using it for scenario development is not clear. I would ask the authors to be more specific about stakeholder input, emphasizing what stakeholders contributed to this effort and what they will do with the products of this research.

Excellent point by the reviewer. In the original submission we cited Lipiec et al., 2018 in which we elaborated on the stakeholder involvement. However, in keeping with the reviewer’s comment on page 2 we now include a few sentences describing the role of stakeholder in our project: The KTAN included members from state, county, and local agencies, nongovernmental organizations, private citizens, researchers, students, and outreach specialists. This stakeholder network was interested in using Envision to evaluate how different adaptation policies and effects of climate change my impact coastal Tillamook County into the future.’  

  1. Modeling experiments emphasize technical aspects of measuring uncertainty. I would ask the authors to explore in the conclusions the implications of these uncertainties for hazard management. How can hazard management professionals make better adaptation decisions in the face of inevitable uncertainty about the climate and policy environment.

We thank Reviewer 3 pointing out this important concern regarding decision making under deep uncertainty. In the conclusion, we explicitly bring this issue up via the comments: ‘Quantifying variability and uncertainty within the Envision framework helped improve the relevance of model results to the project’s stakeholders (Tillamook County Knowledge to Action Network) by allowing for the determination of the relative impact of policy and management decisions on the adaptive capacity of Pacific Northwest coastal communities under a range of future climate scenarios.’ To specifically address the reviewer’s comments we add this sentence to the conclusions: ‘While no alternative presented here is a specific forecast representing the future of Tillamook County, the range of results presented is allowing stakeholders to constrain the deep uncertainty associated with their climate change adaptation planning.’

Reviewer 4 Report

The present manuscript describes an innovative approach for the evaluation of possible outcomes of coastal areas subjected to both climate change and policy decisions of uncertain nature. The authors investigate the role of some of the fundamental climatic and policy drivers, and combined them together to check the possible future scenarios in terms of 1) buildings impacted by either flooding or 2) erosion, and 3) beach accessibility. Although the analysis is undertaken along the US Pacific coast, the approach might be easily applied to similar coastlines worldwide.

The paper structure is suitable and well presents the several investigated aspects, although some points should be carefully addressed.

In the fourth paragraph of P2, something is missing in the last part of the first sentence (“This paper … our study site”). I thus suggest slightly rewording it.

Within the introduction section, I would spend some words on the beach-vulnerability concept and give a general overview of the typical approaches used for this “baseline” description of the coastal region, like the use of coastal vulnerability indexes, described in both less and more recent literature:

  • Gornitz, V. M., Daniels, R. C., White, T. W., & Birdwell, K. R. (1994). The development of a coastal risk assessment database: vulnerability to sea-level rise in the US Southeast. Journal of Coastal Research, 327-338.
  • Boruff, B. J., Emrich, C., & Cutter, S. L. (2005). Erosion hazard vulnerability of US coastal counties. Journal of Coastal research, 21(5 (215)), 932-942.
  • Sekovski, I., Del Río, L., & Armaroli, C. (2020). Development of a coastal vulnerability index using analytical hierarchy process and application to Ravenna province (Italy). Ocean & Coastal Management, 183, 104982.
  • Anfuso, G., Postacchini, M., Di Luccio, D., & Benassai, G. (2021). Coastal Sensitivity/Vulnerability Characterization and Adaptation Strategies: A Review. Journal of Marine Science and Engineering, 9(1), 72.

For the sake of brevity, I would suggest shorter and more incisive titles for some of the subsections (e.g., title of sections 2.1, 2.2, 3.1).

Citations should be checked throughout the text, as round brackets are used at P4 in place of square brackets.

“This hazards” at P4 should be amended (just above eq.2).

All acronyms and symbols should be defined in the text (e.g. MHW, just after eq.3). I suggest adding a list of symbols/abbreviations somewhere in the manuscript.

For a better understanding of the used method, I would add either a sketch or a flowchart of the “Envision” platform in section 2, especially for what concerns the simulation approach related to the community growth and development (section 2.1.2).

Furthermore, I would better explain in section 2.2.1 the way the 33 baseline simulations are built, e.g. which are their bases? why an ascending trend with years?

A legend should be added to the middle panel of Fig.2, so as to reduce a little the caption underneath.

In the caption of Tab.2, the specific section should be recalled, instead of using “previous section”.

In contrast to what assessed in the second paragraph of section 3.1.1, I cannot see any sensitive change in Fig.3c,d provided by the SWH drivers, compared to the baseline condition.

The description of easements and relocating buildings in the first paragraph of P9 should be used to improve Tab.1, i.e. to better distinguish between policies 3 and 4.

At P10, are you sure about 200 buildings? I would say less.

The second sentence of the first paragraph of section 3.1.2 is unclear, please reword.

In the caption of Fig.10, please replace “left”, “middle” and “right” with “a”, “b” and “c”.

The first paragraph of P16 presents some typos, please check. Further, the recalled panel of Fig.11 should be “e”.

In the left panel of Fig.13, what do the authors mean with the policy scenario “hybrid”?

Author Response

The present manuscript describes an innovative approach for the evaluation of possible outcomes of coastal areas subjected to both climate change and policy decisions of uncertain nature. The authors investigate the role of some of the fundamental climatic and policy drivers and combined them together to check the possible future scenarios in terms of 1) buildings impacted by either flooding or 2) erosion, and 3) beach accessibility. Although the analysis is undertaken along the US Pacific coast, the approach might be easily applied to similar coastlines worldwide. The paper structure is suitable and well presents the several investigated aspects, although some points should be carefully addressed.

  • We thank the reviewer for their detailed and thoughtful review.

In the fourth paragraph of P2, something is missing in the last part of the first sentence (“This paper … our study site”). I thus suggest slightly rewording it.

  • Changed to ‘ … adaptation scenarios within our study site - Tillamook County, Oregon (Figure 1)’

Within the introduction section, I would spend some words on the beach-vulnerability concept and give a general overview of the typical approaches used for this “baseline” description of the coastal region, like the use of coastal vulnerability indexes, described in both less and more recent literature:

  • Gornitz, V. M., Daniels, R. C., White, T. W., & Birdwell, K. R. (1994). The development of a coastal risk assessment database: vulnerability to sea-level rise in the US Southeast. Journal of Coastal Research, 327-338.
  • Boruff, B. J., Emrich, C., & Cutter, S. L. (2005). Erosion hazard vulnerability of US coastal counties. Journal of Coastal research, 21(5 (215)), 932-942.
  • Sekovski, I., Del Río, L., & Armaroli, C. (2020). Development of a coastal vulnerability index using analytical hierarchy process and application to Ravenna province (Italy). Ocean & Coastal Management, 183, 104982.
  • Anfuso, G., Postacchini, M., Di Luccio, D., & Benassai, G. (2021). Coastal Sensitivity/Vulnerability Characterization and Adaptation Strategies: A Review. Journal of Marine Science and Engineering, 9(1), 72.
  • We thank the reviewer for these suggestions. However, we note again that our Introduction section already cites 38 papers, many of which are explicitly focused on the potential increased vulnerability of coastal systems due to climate change. While the set of papers listed above are all strong contributions to the literature, we strongly feel that an additional paragraph in our Introduction focused specifically on these semi-quantitative vulnerability indices does not make sense in our manuscript.

For the sake of brevity, I would suggest shorter and more incisive titles for some of the subsections (e.g., title of sections 2.1, 2.2, 3.1).

  • Great suggestion. Section 2.2’s title has been shortened to ‘Evaluating uncertainty.’ At this point we feel that the titles of section 2.1 and 3.1 need to be as explicit as they are but are willing to attempt to shorten if the editor deems this necessary.

Citations should be checked throughout the text, as round brackets are used at P4 in place of square brackets.

  • Good catch. We have fixed these omissions throughout the manuscript.

“This hazards” at P4 should be amended (just above eq.2).

  • This typographical error has been fixed.

All acronyms and symbols should be defined in the text (e.g. MHW, just after eq.3). I suggest adding a list of symbols/abbreviations somewhere in the manuscript.

  • We have defined MHW as mean high water the first time it appears in the manuscript. At this point we have decided against a separate list of symbols/abbreviations due to the limited number in the manuscript.

For a better understanding of the used method, I would add either a sketch or a flowchart of the “Envision” platform in section 2, especially for what concerns the simulation approach related to the community growth and development (section 2.1.2).

  • We worked hard to provide the right level of detail of the various Envision sub-models while not repeating material from earlier publications. For example, Mills et al., 2018 provides a schematic detailing how various modeling components interact. To improve clarity and specifically suggest that the reader refer back to this publication we have added the following sentence: ‘Mills et al. [39] provides a more detailed description of Envision, the coastal hazards sub-models, and the simulation approach related to population growth and development.’

Furthermore, I would better explain in section 2.2.1 the way the 33 baseline simulations are built, e.g. which are their bases? why an ascending trend with years?

  • Our ‘baseline’ climate scenario is characterized by no changes to El Niño frequency, wave climate, or sea level and therefore, from a climate change perspective, does not have an ascending trend with time. However, the reason some of the impact metrics increase with time under this ‘baseline’ scenario is due to two reasons. First, areas experiencing historical shoreline change due to negative sediment budgets are projected to continue to experience erosion, even under no SLR. Second, the growth and development patterns allow for new development close to the hazard zone. To make sure that section 2.2.1 is as clear as possible we modify the last sentence to: ‘Thirty-three synthetic time series of current, or hereinafter baseline (i.e., no changes to sea level, wave climate, or El Niño frequency), climatic conditions were also simulated as a reference case.’

A legend should be added to the middle panel of Fig.2, so as to reduce a little the caption underneath.

  • Figure 2 already has a legend in the left-hand panel and we feel that adding additional legends would make the other panels too busy. We prefer to use the caption to describe the figure in detail.

In the caption of Tab.2, the specific section should be recalled, instead of using “previous section”.

  • Great suggestion. We have simply called out Table 1 explicitly now in Table 2’s caption.

In contrast to what assessed in the second paragraph of section 3.1.1, I cannot see any sensitive change in Fig.3c,d provided by the SWH drivers, compared to the baseline condition.

  • This paragraph is meant to highlight that SLR has the most important influence on the number of buildings impacted by flooding. However, we also want to point out that the significant wave height climate has more of an influence, even if relatively minor, than modifying the frequency of major El Niño events. To address the reviewers comment, this paragraph has been modified to include this sentence: ‘While the frequency of major El Niño events did not significantly shift metric values from the baseline (Figure 3e,f), changes to the wave climate did have an influence on flooding (Figure 3c,d), albeit relatively minor, particularly in the latter half of the century.’

The description of easements and relocating buildings in the first paragraph of P9 should be used to improve Tab.1, i.e. to better distinguish between policies 3 and 4.

  • In this paragraph we have now called out the three policies that make a different to this metric by their Policy number as shown in Table 1. We have also added this sentence to further clarify the differences between policies 3 and 4: ‘The easement policy (Policy 3) completely removed structures from areas impacted by hazards while the relocate policy (Policy 4) only delayed the ultimate exposure to erosion by simply moving existing structures within an existing parcel.’

At P10, are you sure about 200 buildings? I would say less.

  • Agreed, and good catch. We have changed this to 100 buildings.

The second sentence of the first paragraph of section 3.1.2 is unclear, please reword.

  • We have split this sentence up into two sentences in an attempt to clarify and have added comments directing the reader towards aspects of Figure 10 that help to explain the concept.

In the caption of Fig.10, please replace “left”, “middle” and “right” with “a”, “b” and “c”.

  • Done

The first paragraph of P16 presents some typos, please check. Further, the recalled panel of Fig.11 should be “e”.

  • We now refer to the appropriate figure 11 panels in this paragraph

In the left panel of Fig.13, what do the authors mean with the policy scenario “hybrid”?

  • Thanks for catching this. We uploaded the wrong version of Figure 13. We have now included the correct version of Figure 13.

Round 2

Reviewer 4 Report

I am glad that most of the comments have been addressed by the authors. However, I believe that tackling some of the other comments, not addressed by the authors, would well improve their work.

  • I strongly believe that speaking about semi-quantitative indices is important to give a different (also valid and robust) footprint to the discussed issue. If an additional (brief) paragraph is important for the state of the art of the paper, it cannot be a matter of number of cited papers!
  • I am convinced that a list of symbols does not require so much time to be produced, rather it can give an added value to the paper.
  • The addition of a simple graph/flowchart (perhaps similar to one already included on Mills et al., 2018) to better focus the reader’s attention on the Envision platform would be paramount for the paper clarity. In my opinion, the citation is useful if one wants to go into the model details, but a graph included here would rapidly clarify possible reader’s doubts.
  • In figure 2, the legend in the left panel perhaps leads to confusion, because it suggests that such legend is shared by both left and middle panels, although it is not! A dedicated legend is thus required for the middle panel, this also identifying the meaning of each line much better than the description in the caption.

Author Response

I am glad that most of the comments have been addressed by the authors. However, I believe that tackling some of the other comments, not addressed by the authors, would well improve their work.

  • I strongly believe that speaking about semi-quantitative indices is important to give a different (also valid and robust) footprint to the discussed issue. If an additional (brief) paragraph is important for the state of the art of the paper, it cannot be a matter of number of cited papers!
  • We have added several sentences to the 2nd paragraph of the Introduction to introduce these semi-quantitative indices and have added 4 citations to the manuscript.
  • I am convinced that a list of symbols does not require so much time to be produced, rather it can give an added value to the paper.
  • A list of symbols has been added to the paper as Appendix B.
  • The addition of a simple graph/flowchart (perhaps similar to one already included on Mills et al., 2018) to better focus the reader’s attention on the Envision platform would be paramount for the paper clarity. In my opinion, the citation is useful if one wants to go into the model details, but a graph included here would rapidly clarify possible reader’s doubts.
  • Figure 2 is now a flow schematic that describes the modeling using the Envision platform.
  • In figure 2, the legend in the left panel perhaps leads to confusion, because it suggests that such legend is shared by both left and middle panels, although it is not! A dedicated legend is thus required for the middle panel, this also identifying the meaning of each line much better than the description in the caption.
  • The legend of former figure 2 (now figure 3) has ben modified to include legends on all 3 panels.
